# Robust Incremental Neural Semantic Graph Parsing

## Abstract

Parsing sentences to linguistically-expressive semantic representations is a key goal of Natural Language Processing. Yet statistical parsing has focussed almost exclusively on bilexical dependencies or domain-specific logical forms. We propose a neural encoder-decoder transition-based parser which is the first full-coverage semantic graph parser for Minimal Recursion Semantics (MRS). The model architecture uses stack-based embedding features, predicting graphs jointly with unlexicalized predicates and their token alignments. Our parser is more accurate than attention-based baselines on MRS, and on an additional Abstract Meaning Representation (AMR) benchmark, and GPU batch processing makes it an order of magnitude faster than a high-precision grammar-based parser. Further, the $86.69\%$ Smatch score of our MRS parser is higher than the upper-bound on AMR parsing, making MRS an attractive choice as a semantic representation.

## 1 Introduction

An important goal of Natural Language Understanding (NLU) is to parse sentences to structured, interpretable meaning representations that can be used for query execution, inference and reasoning. While it has recently been shown that end-to-end models outperform traditional pipeline approaches using syntactic or semantic parsers on many NLU tasks, those parses were frequently relatively shallow, e.g. restricted to projective bilexical dependencies.

In this paper we focus on the robust parsing of linguistically deep semantic representations. The main representation that we use is Minimal Recursion Semantics (MRS) (Copestake et al., 1995, 2005), which serves as the semantic representation of the English Resource Grammar (ERG) (Flickinger, 2000). The only previous approach to parsing and disambiguating full MRS structures (as opposed to bilexical semantic graphs derived from, but simplifying MRS) were based on the ERG (Toutanova et al., 2005); this approach has high precision but incomplete coverage.

Our main contribution is to develop a fast and robust parser for full MRS-based semantic graphs. We exploit the power of global conditioning enabled by deep learning to predict linguistically deep graphs incrementally. The model does not have access to the underlying ERG or syntactic structures from which the MRS analyses were originally derived. We develop parsers for two graph-based conversions of MRS, Elementary Dependency Structure (EDS) (Oepen and Lønning, 2006) and Dependency MRS (DMRS) (Copestake, 2009), of which the latter is inter-convertible with MRS.

Abstract Meaning Representation (AMR) (Banarescu et al., 2013) is a graph-based semantic representation with similar goals to that of MRS. Apart from differences in the choice of which linguistic phenomena are annotated, MRS is a compositional representation explicitly coupled with the syntactic structure of the sentence, while AMR does not assume compositionality or alignment with the sentence structure. AMR parsing has recently received a lot of attention, but the size of the available training data is still relatively small, and inter-annotator agreement has been shown to be relatively low, placing on upper bound of $83\%$ F1 on the expected parser performance. We apply our model to AMR parsing by introducing struc-

ture (alignments and distinguishing between lexical and non-lexical concepts) that is present explicitly in MRS but not in AMR.

Parsers based on RNNs have achieved state-of-the-art performance for dependency parsing (Dyer et al., 2015; Kiperwasser and Goldberg, 2016) and constituency parsing (Vinyals et al., 2015b; Dyer et al., 2016; Cross and Huang, 2016b). However, one of the main advantages of deep learning is the ability to make predictions conditioned on the unbounded contexts encoded with RNNs; this enables us to predict more complex structures than have been possible previously, without increasing algorithmic complexity. One of the main reasons for the prevalence of dependency parsing, including semantic dependency parsing (Ivanova et al., 2013), is that it can be performed with efficient and well-understood algorithms. Therefore deep learning gives us the opportunity to perform robust, linguistically deep parsing.

Our parser is a transition-based model for parsing semantic graphs. However, instead of generating arcs over an ordered, fixed set of nodes (the words in the sentence), we generate the nodes, including their labels and alignments to the input tokens, jointly with the transition actions. We use a variant of the arc-eager transition-system that is able to parse graphs and non-planar dependencies. The sentence is encoded with a bidirectional RNN. The transition sequence, seen as a graph linearization, can be predicted with any encoder-decoder model, but we show that using hard attention, predicting the alignment with a pointer network and conditioning explicitly on stack-based features improves performance. In order to deal with data sparsity candidate lemmas are predicted as a preprocessing step, so that the RNN decoder predicts unlexicalized predicates.

We evaluate our parser on DMRS, EDS and AMR graphs. We show that our model architecture improves performance from $79.68\%$ to $84.16\%$ F1 over an attention-based encoder-decoder baseline. Although the model is less accurate that a high-precision grammar-based parser on a test set of sentences parsable by that grammar, our model is an order of magnitude faster due to incremental prediction and a GPU batch processing implementation of the transition system. On AMR parsing our model obtains $60.11\%$ Smatch, an improvement of $8\%$ over an existing neural AMR parser.

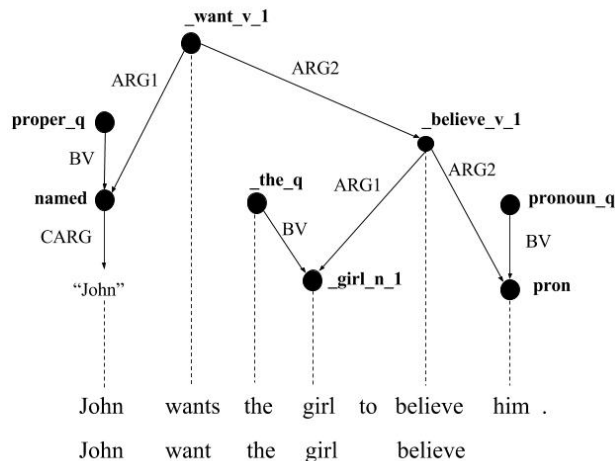

Figure 1: Deep semantic representation of the sentence "John wants the girl to believe him." The graph is based on the Elementary Dependency Structure (EDS) representation of Minimal Recursion Semantics (MRS). The sentence's tokens and lemmas are given with node to token alignments.

## 2 Deep Meaning Representations

We define a common framework for semantic graphs, in which we can place both MRS-based graph representations (DMRS and EDS) and AMR. In this framework sentence meaning is represented with rooted, labelled, connected, directed graphs. An example graph is visualized in Figure 1. Node labels are referred to as *predicates* (*concepts* in AMR) and edge labels as *arguments* (AMR *relations*). In addition, a special class of node modifiers, *constants*, are used to denote the string values of named entities and numbers (including date and time expressions). Every node is aligned to a token or a continuous span of tokens in the sentence the graph corresponds to.

Minimal Recursion Semantics (MRS) is a framework for computational semantics that can be used for parsing or generation (Copestake et al., 2005). The main units of MRS are *elementary predications* (EPs). An EP consists of a relation (referred to as a *predicate*), usually corresponding to a single lexeme, and its arguments. Quantification is expressed by relations, not logical operations like $\exists$ or $\forall$. One of the distinguishing characteristics of MRS its support for scope underspecification; multiple scope-resolved logical representations can be derived from one MRS structure. MRS was designed to be integrated

with feature-based grammars, and has been implemented in English within the framework of Head-driven Phrase Structure Grammar (HPSG) (Pollard and Sag, 1994) in the English Resource Grammar (ERG) (Flickinger, 2000).

MRS can be converted without loss to variable-free dependency graphs, called Dependency MRS (DMRS) (Copestake, 2009; Copestake et al., 2016). A similar graph-based conversion is Elementary Dependency Structures (EDS) (Oepen and Lønning, 2006), which drops the scope-underspecification machinery, primarily simplifying edge labels. Figure 1 illustrates an EDS graph.

MRS makes an explicit the distinction between lexical and non-lexical predicates (lexical predicates are prefixed by an underscore). Lexical predicates consist of a lemma followed by a coarse part-of-speech tag and an optional sense label. Predicates absent from the ERG lexicon are represented by their surface forms, POS tags and an unknown sense label. All predicates are annotated with an alignment to the character-span of the (untokenized) input sentence. We convert the character-level spans given by MRS to token-level spans. Lexical predicates usually align with the span of the token(s) they represent, while non-lexical predicates can span longer segments. In full MRS predicates are annotated with a set of morphosyntactic features consisting of attribute-value pairs, but we do not currently model these features.

AMR (Banarescu et al., 2013) graphs can be represented in the same framework, despite a number of linguistic differences with MRS. However, information annotated explicitly in MRS is considered as latent in AMR. This include alignments, as well as distinguishing between lexical and non-lexical concepts. AMR predicates are based on PropBank (Palmer et al., 2005), annotated as lemmas plus sense labels, but they form only a subset of concepts. Other concepts are either English words or special keywords, and can correspond to overt lexemes in some cases but not others.

## 3 Incremental Graph Parsing

We parse sentences to their meaning representations by incrementally predicting semantic graphs together with their alignments. Let $\mathbf{e} = e_1, e_2, \ldots, e_I$ be a tokenized English sentence, $\mathbf{t} = t_1, t_2, \ldots, t_J$ a sequential representation of its graph derivation and $\mathbf{a} = a_1, a_2, \ldots, a_J$ an align-

```
:root( <1> _v_1
  :ARG1( <0> named_CARG
    :BV-of( <0> proper_q ) )
  :ARG2 <5> _v_1
    :ARG1( <3> _n_1
      :BV-of ( <2> _q ) )
    :ARG2( <5> pron
      :BV-of ( <5> pronoun_q ) ) )
```

Figure 2: A top-down linearization of the EDS graph in Figure 1, using unlexicalized predicates.

ment sequence consisting of integers in the range $1, \ldots, I$. We model the conditional distribution $p(\mathbf{t}, \mathbf{a}|\mathbf{e})$ which decomposes as

$$\prod_{j=1}^{J} p(a_j|(\mathbf{a}, \mathbf{t})_{1:j-1}, \mathbf{e})p(t_j|\mathbf{a}_{1:j}, \mathbf{t}_{1:j-1}, \mathbf{e}).$$

We also predict the end-of-span alignments as a seperate sequence $\mathbf{a}^{(e)}$.

### 3.1 Top-down linearization

We now consider how to linearize the semantic graphs, before defining the neural models to parameterize the parser in section 4. The first approach is to linearize a graph as the pre-order traversal of its spanning tree, starting at a designated root node (see Figure 2). Variants of this approach has been proposed for neural constituency parsing (Vinyals et al., 2015b), logical form prediction (Dong and Lapata, 2016; Jia and Liang, 2016) and AMR parsing (Barzdins and Gosko, 2016; Peng et al., 2017).

In the linearization, labels of edges whose direction are reversed in the spanning tree are marked by adding -of. Edges not included in the spanning tree, referred to as *reentrancies*, are represented with special with edges whose dependents are dummy nodes pointing back to the original nodes. Our potentially lossy representation represents the edge by repeating the label and alignment of the depenen node, which is recovered heuristically. The alignment does not influence the order of the nodes in this linearization.

### 3.2 Arc-eager parsing

Figure 1 shows that the semantic graphs we work with can also be interpreted as dependency graphs, as nodes are aligned to sentence tokens. An approach to predicting dependency graphs incrementally that has been used extensively is transition-

based parsing (Nivre, 2008). We apply a variant of the arc-eager transition system that has been proposed for graph (as opposed to tree) parsing (Titov et al., 2009; Gómez-Rodríguez and Nivre, 2010) to derive a transition-based parser for deep semantic graphs. In dependency parsing the sentence tokens also act as nodes in the graph, but here we need to generate the nodes incrementally as the transition-system proceeds, conditioning the generation on the given sentence. Damonte et al. (2016) proposed an arc-eager AMR parser, but the definition of their system is more narrowly defined for AMR graphs.

The transition system consists of a *stack* of graph nodes being processed and a *buffer*, holding a single node at a time. The transition actions are *shift*, *left-arc*, *right-arc* and *reduce*. An example transition sequence is given in Figure 3, together with the stack and buffer after each step. The shift transition moves the element on the buffer to the top of the stack, and generates a predicate and its alignment as the next node on the buffer. Left-arc and right-arc actions add labeled arcs between the buffer and stack top (for DMRS a transition for undirected arcs is included), but do not change the state of the stack or buffer. Finally, reduce pops the top element from the stack, and predicts its end-of-span alignment if included in the representation. To predict non-planar arcs, we add another transition, which we call *cross-arc*, which first predicts the stack index of a node which is not on top of the stack, adding an arc between the head of the buffer and that node. Another special transition designates the buffer node as the root.

To derive an oracle for this transition system, it is necessary to determine the order in which the nodes are generated. We consider two different orderings. The first is that of an in-order traversal of the spanning tree, where the node order is determined by the alignment. This leads to a linearization where the only non-planar arcs are reentrancies. The second is to make the ordering non-decreasing with respect to the alignments, while for nodes with the same alignment following the in-order ordering. In an arc-eager oracle arcs are added greedily, while a reduce action can either be performed as soon as the stack top node has been connected to all its dependents, or delayed until it has to reduce to allow the correct parse tree to be formed. In our model the oracle delays reduce, where possible, until the end alignment of

the stack top node spans the node on the buffer. As the span end alignments often cover phrases that they head (e.g. for quantifiers) this gives a natural interpretation to predicting the span end together with the reduce action.

### 3.3 Lemma prediction

Lexical predicates in MRS consist of a lemma followed by a sense label. Therefore if we can predict the alignment of a graph node as well as the lemma of the aligned word, the prediction required by the decoder can be simplified to only predicting the sense label. We extract a dictionary mapping words to lemmas from the ERG lexicon, which map words to their possible predicates. In combination with a lemmatizer, this can be used to predict a candidate lemma for each token. The same approach is applied to predict constants, together with additional normalizations, e.g. to map numbers to digit strings.

We use the Stanford CoreNLP toolkit (Manning et al., 2014) to tokenize and lemmatize sentences, and tag tokens with the Stanford Named Entity Recognizer (Finkel et al., 2005). For AMR parsing there is not annotated distinction between lexical and non-lexical tokens, so we automatically-obtained alignments and the graph structure to classifiy concepts as lexical or non-lexical. The lexicon is restricted to Propbank (Palmer et al., 2005) predicates, so for other concepts we extract a lexicon from the training data.

## 4 Encoder-Decoder Models

### 4.1 Sentence encoder

The sentence **e** is encoded with a bidirectional RNN. We use a standard LSTM architecture without peephole connections (Jozefowicz et al., 2015). For every token $e$ we embed its word, POS tag and named entity (NE) tag as vectors $x_w$, $x_t$ and $x_n$, respectively.

The embeddings are concatenated and passed through a linear transformation,

$$g(e) = W^{(x)}[x_w; x_t; x_n] + b^x$$

such that $g(e)$ has the same dimension as the LSTM. We don't embed word lemmas separately for the unlexicalized representations. Each input position $i$ is represented by a hidden state $h_i$, which is the concatenation of its forward and backward LSTM state vectors.

| Action | Stack | Buffer | Arc added |
|---|---|---|---|
| init(0, named_CARG) | [ ] | (0, 0, named_CARG) | - |
| sh(0, proper_q) | [(0, 0, named_CARG)] | (1, 0, proper_q) | - |
| la(BV) | [(0, 0, named_CARG)] | (1, 0, proper_q) | (1, BV, 0) |
| sh(1, _v_1) | [(0, 0, named_CARG), (1, 0, proper_q)] | (2, 1, _v_1) | - |
| re | [(0, 0, named_CARG)] | (2, 1, _v_1) | - |
| la(ARG1) | [(0, 0, named_CARG)] | (2, 1, _v_1) | (2, ARG1, 0) |

Figure 3: First part of the transition sequence for parsing the graph in Figure 1. The transitions are shift (sh), reduce (re), left arc (la) and right arc (ra). The action taken at each step is given, along with the state of the stack and buffer after the action is applied, and any arcs added. Shift transitions generate the alignment and predicate of the next graph node. Items on the stack and buffer have the form (*node index, alignment, predicate label*), and arcs are of the form (*head index, argument label, dependent index*).

## 4.2 Hard attention decoder

We model the alignment of graph nodes to sentence tokens, $\mathbf{a}$, as a random variable. For the arc-eager model, $a_j$ corresponds to the alignment of the node of the buffer after action $t_j$ is executed. The distribution of $t_j$ is over all transitions and predicate predictions (for shifts), predicted with a single softmax.

Let $s_j$ be the RNN decoder hidden state at output position $j$. We initialize $s_0$ with the final state of the backward encoder LSTM. The alignment is predicted with a pointer network (Vinyals et al., 2015a).

The logits are computed with an MLP scoring the decoder hidden state against each of the encoder hidden states (for $i = 1, \ldots, I$),

$$u_j^i = v^T \tanh(W^{(1)} h_i + W^{(2)} s_j).$$

The alignment distribution is then estimated by

$$p(a_j = i | \mathbf{a}_{1:j-1}, \mathbf{t}_{1:j-1}, \mathbf{e}) = \text{softmax}(u_j^i).$$

To predict the next transition $t_i$, the output vector is conditioned on the encoder state vector $h_{a_j}$, corresponding to the alignment:

$$o_j = W^{(3)} s_j + W^{(4)} h_{a_j}$$
$$v_j = R^{(d)} o_j + b^{(d)},$$

where $R^{(d)}$ and $b^{(d)}$ are the output representation matrix and bias vector, respectively.

The transition distribution is then given by

$$p(t_j | \mathbf{a}_{1:j}, \mathbf{t}_{1:j-1}, \mathbf{e}) = \text{softmax}(v_j).$$

Let $v(t)$ be the embedding of decoder symbol $t$. The RNN state at the next time-step is computed as

$$d_{j+1} = W^{(5)} v(t_j) + W^{(6)} h_{a_j}$$
$$s_{j+1} = RNN(d_{j+1}, s_j).$$

The end-of-span alignment $a_j^{(e)}$ for MRS-based graphs is predicted with another pointer network. The end alignment of a token is predicted only when a node is reduced from the stack, therefore this alignment is not observed at each time-step; it is also not fed back into the model.

The hard attention approach, based on supervised alignments, can be contrasted to soft attention, which learns to attend over the input without supervision. The attention is computed as with hard attention, as $\alpha_j^i = \text{softmax}(u_j^i)$. However instead of making a hard selection, a weighted average over the encoder vectors is computed as $q_j = \sum_{i=1}^{i=I} \alpha_j^i h_i$. This vector is used instead of $h_{a_j}$ for prediction and feeding to the next time-step.

## 4.3 Stack-based model

We extend the hard attention model to include features based on the transition system stack. Elements on the stack can be represented by the encoder biLSTM representations corresponding to the tokens they are aligned to. We include vectors for the top of the stack and the buffer, the latter which is predicted by the hard attention. This approach is similar to the features proposed by Kiperwasser and Goldberg (2016) and Cross and Huang (2016a) for dependency parsing. The layer that computes the output vector is extended to

$$o_j = W^{(3)} s_j + W^{(4)} h_{a_j} + W^{(7)} h_{\text{st}_0}$$

where $\text{st}_0$ is the sentence alignment index of the top element on the buffer. The input layer to the

next RNN time-step is similarly extended to

$$d_{j+1} = W^{(5)}v(t_j) + W^{(6)}h_{\text{buf}} + W^{(8)}h_{\text{st}_0},$$

where `buf` is the buffer alignment after $t_j$ is executed.

Our implementation of the stack-based model enables batch processing in static computation graphs, similar to Bowman et al. (2016). We maintain a stack of alignment indexes for each element in the batch, which is updated inside the computation graph after each parsing action.

We perform greedy decoding. For the stack-based model we ensure that if the stack is empty, the next transition predicted has to be shift. For the other models we ensure that the output is well-formed during post-processing by robustly skipping over out-of-place symbols or inserting missing ones.

## 5 Experiments

### 5.1 Data

DeepBank (Flickinger et al., 2012) is HPSG and MRS annotation of the Penn Treebank Wall Street Journal (WSJ) corpus text, developed following an approach known as dynamic treebanking (Oepen et al., 2004), that couples treebank annotation with grammar development, in the case of the ERG. This approach has been shown to lead to high inter-annotator agreement: $0.94$ against $0.71$ for AMR (Bender et al., 2015). Parses are only provided for sentences for which the ERG has an analysis acceptable to the annotator – this means that we cannot evaluate parsing accuracy for sentences which the ERG cannot parse (approximately $15\%$ of the original corpus).

We use Deepbank version $1.1$, corresponding to ERG `1214`[1], following the suggested split of sections 0 to 19 as training data data, 20 for development and 21 for testing. The gold-annotated training data consists of 35,315 sentences. We use the pyDelphin library[2] and software provided with the ERG to extract DMRS and EDS graphs.

For AMR parsing we use LDC2015E86, the dataset released for the SemEval 2016 AMR parsing Shared Task (May, 2016). This data includes newswire, weblog and discussion forum text. The training set has 16,144 sentences. We obtain align-

ments using the rule-based JAMR aligner (Flanigan et al., 2014).

### 5.2 Evaluation

Dridan and Oepen (2011) proposed an evaluation metric called Elementary Dependency Matching (EDM) for MRS graphs. EDM computes the F1-scores of tuples of predicates and arguments. A predicate tuples consist of the label and character span of a predicate, while an argument tuple consist of the character spans of the head and dependent of the relation, together with the argument label. In order to tolerate tokenization differences with respect to punctuation, we allow span pairs whose ends differ by 1 character to be matched.

EDM can be contrasted with the Smatch metric (Cai and Knight, 2013) for evaluating AMR graphs. This evaluation does not rely on sentence alignments; instead it performs inference over graph alignments to estimate the maximum F1-score obtainable from a 1-1 matching between the predicted and gold graph nodes.

### 5.3 Model Setup

Our models are implemented in TensorFlow (Abadi et al., 2015). For training we use Adam (Kingma and Ba, 2015) with learning rate $0.01$ and batch-size $64$. Gradients norms are clipped to $5.0$ (Pascanu et al., 2013). We use single-layer LSTMs with dropout of $0.3$ (tuned on the development set) on input and output connections. We use encoder and decoder embeddings of size $256$, and POS and NE tag embeddings of size $32$, For DMRS and EDS graphs the hidden units size is set to $256$, for AMR it is $128$. This configuration, found using grid search and heuristic search within the range of models that fit into a single GPU, gave the best performance on the development set under multiple graph linearizations. Encoder word embeddings are initialized (in the first 100 dimensions) with pre-trained order-sensitive embeddings (Ling et al., 2015). Singletons in the encoder input is replaced with an unknown word symbol with probability $0.5$ for each iteration.

### 5.4 MRS parsing results

We compare different linearizations and model architectures for parsing DMRS on the development data, showing that our approach is more accurate than baseline neural approaches. We report

---

[1] http://svn.delph-in.net/erg/tags/1214/
[2] https://github.com/delph-in/pydelphin

| Model | EDM | EDM-Pred | EDM-Arg |
|---|---|---|---|
| TD lex | 81.44 | 85.20 | 76.87 |
| TD unlex | 81.72 | 85.59 | 77.04 |
| AE lex | 81.35 | 85.79 | 76.02 |
| AE delex | 82.56 | 86.76 | 77.54 |

Table 1: DMRS development set results for attention-based encoder-decoder models with alignments encoded in the linearization, for top-down (TD) and arc-eager (AE) linearizations, and lexicalized and unlexicalized predicate prediction.

| Model | EDM | EDM-Pred | EDM-Arg |
|---|---|---|---|
| TD soft | 81.53 | 85.32 | 76.94 |
| TD hard | 82.75 | 86.37 | 78.37 |
| AE hard | 84.65 | 87.77 | 80.85 |
| AE stack | 85.28 | 88.38 | 81.51 |

Table 2: DMRS development set results of pointer-augmented encoder-decoder models with hard and soft attention architectures.

EDM F1 scores, as well as EDM scores for predicate (EDM-Pred) and argument (EDM-Arg) prediction.

First we report results using a vanilla encoder-decoder model (Table 1). We compare the top-down and arc-eager linearizations, as well as the effect of delexicalizing the predicates (factorizing lemmas out of of the predicate labels and predicting them separately.) In both cases constants are predicted with a dictionary lookup based on the predicted spans; for predicates not in the lexicon an unknown-label is predicted and the words and POS tags are recovered during post-processing.

The arc-eager unlexicalized linearization gives the best performance, down linearization, even though the model has to learn to model the transition stack inside the recurrent states without any supervision of the semantics of transition actions. The unlexicalized models are more accurate, mostly due to their ability to generalize to sparse or unseen predicates occurring in the lexicon. For the arc-eager representation, the oracle EDM is $99\%$ for the lexicalized representation and $98.06\%$ for the delexicalized representation. The remaining errors are due to discrepancies between the tokenization used by our system and the ERG tokenization. The unlexicalized models are also faster to train, as the decoder's output vocabulary is much smaller, reducing the expense of computing softmaxes over large vocabularies.

| Model | Att-RNN | AE-RNN | ACE |
|---|---|---|---|
| EDM | 79.68 | 84.16 | 89.64 |
| Smatch | 85.28 | 86.69 | 93.50 |
| EDM-Start | 84.44 | 87.81 | 91.91 |
| EDM-Pred | 83.36 | 87.54 | 92.08 |
| EDM-Arg | 75.16 | 80.10 | 86.77 |
| EDM-Start-Arg | 80.93 | 85.61 | 89.28 |

Table 3: DMRS parsing test set results.

Next we consider models that predict the alignments with pointer networks, contrasting soft and hard attention models (Table 2). The results show that the arc-eager models performs better than those based on top-down representation. For the arc-eager model we use hard attention, due to the natural interpretation of the alignment prediction corresponding to the transition system. The arc-eager stack-based architecture improves further over the model that purely relies on the hard attention.

We compared the effect of different orderings of the predicates for the arc-eager model: The monotone ordering (with non-decreasing alignments) performs $0.44$ EDM better that the in-order ordering, despite having to parse more non-planar dependencies. We also compare the predicate prediction against a hard attention model that only predicts predicates (in monotone order) together with their start spans. This model obtains $91.36\%$ F1 on predicates together with their start spans with the delexicalized model, compared to $88.22\%$ for lexicalized predicates and $91.65\%$ for the full parsing model.

We present test set results for various metrics in Table 3. We compare the performance of our neural baseline and stack-based decoders with the ACE[3] ERG-based parser. Another approach to robust MRS parsing has previously been proposed (Zhang et al., 2014), but no comparable results or implementation is available.

Despite the promising performance of the model there is still a gap between the accuracy of our parser and ACE. One reason for this is that the test set sentences will arguably be easier for ACE to parse as their choice was restricted by the grammar that ACE uses. EDM metrics excluding end-span prediction (-Start) show that our parser has relatively more difficulty in parsing end-span predictions than the grammar-based parser.

---

[3] http://sweaglesw.org/linguistics/ace/

| Model | ArcEager | ACE |
|---|---|---|
| EDM | 85.48 | 89.58 |
| Smatch | 86.50 | 93.52 |
| EDM-Pred | 88.14 | 91.82 |
| EDM-Arg | 82.20 | 86.92 |

Table 4: EDS parsing test set results.

| Model | Concept F1 | Smatch |
|---|---|---|
| TD no point | 65.55 | 57.95 |
| TD soft | 66.44 | 59.36 |
| TD soft delex | 67.07 | 59.88 |
| AE hard delex | 72.86 | 59.83 |
| AE stack delex | 73.69 | 61.21 |

Table 5: Development set results for AMR parsing. All the models are pointer-based, except where indicated otherwise.

We also evaluate the speed of our model compared with ACE. For the unbatched version of our model, the stack-based parser parses 41.63 tokens per second, while the baseline with linearized spans parses 31.65 tokens per second. The batched (stack-based) implementation parses 529.42 tokens per second, using a batch size of 128. In comparison, the setting of ACE for which we reported accuracies parses 7.47 tokens per second. By restricting the memory usage of ACE, which restricts its coverage, we see that ACE can parse 11.07 tokens per second at $87.7\%$ coverage, and 15.11 tokens per second at $77.8\%$ coverage.

Finally we report results for parsing EDS (Table 4) The EDS parsing task is slightly simpler than DMRS, due to the absence of label and structural annotations that allows for the recovery of MRS, including scope underspecification. These additional labels are harder to predict without access to a grammar.

### 5.5 AMR parsing

We now apply the same approach to AMR parsing. Results on the development set are given in Table 5. The arc-eager-based models again give better performance, mainly due to improved concept prediction accuracy. However, concept prediction remains a weakness of the model; Damonte et al. (2016) reports that state-of-the-art AMR parsers score between $79\%$ and $83\%$ on concept prediction. In contrast to DMRS parsing, the in-order linearization gives better results than the monotone one. We hypothesize that this is due to the

| Model | LDC |
|---|---|
| JAMR | 56 |
| Wang et al. (2016) | 66.54 |
| Damonte et al. (2016) | 64 |
| Peng and Gildea (2016) | 55 |
| Peng et al. (2017) | 52 |
| Barzdins and Gosko (2016) | 43.3 |
| TD seq2seq | 56.56 |
| AE pointer stack | 60.11 |

Table 6: AMR parsing test set results (Smatch F1 scores). Published results follow the number of decimals which were reported.

non-compositional nature of AMR.

We show that even our neural baseline outperforms existing results for neural AMR parsing. The arc-eager model does not perform better than the top-down linearization, which we hypothesize is due to noise in the automatically-obtained alignments.

We report test set results on LDC2015E86 test set (Table 6). Our best neural model outperforms the baseline JAMR parser (Flanigan et al., 2014), but still lags behind the performance of state-of-the-art AMR parsers such as CAMR (Wang et al., 2016). However these models make extensive use of external resources, including syntax trees and semantic role labelling. We see that our attention-based encoder-decoder model already performs better than previous sequence-to-sequence AMR parsers (Barzdins and Gosko, 2016; Peng et al., 2017), and the arc-eager model boost accuracy further. Our model also outperforms a Synchronous Hyperedge Replacement Grammar model (Peng and Gildea, 2016) which is comparable as it does not make extensive use of external resources.

## 6 Conclusion

In this paper we advance the state of parsing by employing deep learning techniques to parse sentence to linguistically expressive semantic representations that have not previously been parsed in an end-to-end fashion. We presented a robust, wide-coverage parser for MRS that is faster than existing parsers and amenable to batch processing. We believe that there are many future avenues to explore to further increase the accuracy of such parsers, including different training objectives, more structured architectures and semi-supervised learning.

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
