# Peer review of "Robust Incremental Neural Semantic Graph Parsing"

_ACL 2017 — decision unknown_

[Official Review · Reviewer 1 · rating 5 · confidence 4]
soundness 3 · originality 4 · clarity 4 · impact 4 · substance 5 · appropriateness 5 · meaningful comparison 4 · presentation format Oral Presentation

- Strengths:
The paper proposes an end-to-end neural model for semantic graph parsing,
based on a well-designed transition system. 
The work is interesting, learning
semantic representations of DMRS, which is capable of resolving semantics
such as scope underspecification. This work shows a new scheme for
computational semantics, benefiting from an end-to-end transition-based
incremental framework, which resolves the parsing with low cost.

- Weaknesses:
  My major concern is that the paper only gives a very common introduction for
the
definition of DMRS and EP, and the example even makes me a little confused
because I cannot see anything special for DMRS. The description can be a little
more detailed, I think. However, upon the space limitation, it is
understandable. The same problem exists for the transition system of the
parsing model. If I do not have any background of MRS and EP, I can hardly
learn something from the paper, just seeing that this paper is very good.

- General Discussion:
  Overall, this paper is very interesting to me. I like the DMRS for semantic
parsing very much and like the paper very much. Hope that the open-source codes
and datasets can make this line of research being a hot topic.